# Impacts of elevated anthropogenic emissions on physicochemical characteristics of BC-containing particles over the Tibetan Plateau

Jinbo Wang[1,2], Jiaping Wang[1,2,3*], Yuxuan Zhang[1,2,3,4], Tengyu Liu[1,2,3], Xuguang Chi[1,2,3], Xin Huang[1,2], Dafeng Ge[1,2], Shiyi Lai[1,2], Caijun Zhu[1,2], Lei Wang[1,2,3], Qiaozhi Zha[1,2,3], Ximeng Qi[1,2,3], Wei Nie[1,2,3], Congbin Fu[1,2,3] and Aijun Ding[1,2,3]

[1]Joint International Research Laboratory of Atmospheric and Earth System Sciences, School of Atmospheric Sciences, Nanjing University, Nanjing, 210023, China.

[2]Jiangsu Provincial Collaborative Innovation Center of Climate Change, Nanjing, 210023, China.

[3]National Observation and Research Station for Atmospheric Processes and Environmental Change in Yangtze River Delta, Nanjing, 210023, China.

[4]Key Laboratory of Atmospheric Environment and Extreme Meteorology, Institute of Atmospheric Physics, Chinese Academy of Sciences, Beijing, 100029, China.

*Correspondence to*: Jiaping Wang (wangjp@nju.edu.cn)

**Abstract.**

Black carbon (BC) in the Tibetan Plateau (TP) region has distinct climate effect, which strongly depends on its mixing state. The aging processes of BC in TP are subject to emissions from various regions, resulting in considerable variability of its mixing state and physicochemical properties. However, the mechanism and magnitude of this effect are not yet clear. In this study, filed observations on physicochemical properties of BC-containing particles ($PM_{BC}$) were conducted in the northeast (Xihai) and southeast (Lulang) regions of the TP to investigate the impacts of transported emissions from lower-altitude areas on BC characteristics in the TP. Large spatial discrepancies were found in the chemical composition of $PM_{BC}$. Both sites showed higher concentrations of $PM_{BC}$ when they were affected by transported airmasses outside the TP, but with diverse chemical composition. Source apportionment for organic aerosol (OA) suggested that primary OA in the northeastern TP was attributed to hydrocarbon OA (HOA) from anthropogenic emissions, while it was dominated by biomass burning OA (BBOA) in the southeastern TP. Regarding secondary aerosol, a marked enhancement in nitrate fraction was observed on aged BC coating in Xihai when the airmasses were brought by updrafts and easterly winds from lower-altitude areas. With the development of boundary layer, the enhanced turbulent mixing promoted the elevation of anthropogenic pollutants. In contrast to Xihai, the thickly coated BC in Lulang was mainly caused by elevation and transportation of biomass burning plume from the South Asia, showing a large contribution of secondary organic aerosol (SOA). The distinct transported emissions lead to substantial variations of both chemical composition and light absorption ability of BC across the TP. The thicker coating and higher mass absorption cross-section (MAC) of $PM_{BC}$ in airmasses elevated from lower-altitude regions reveals the promoted BC aging processes and their impacts on the mixing state and light absorption of BC in TP. These findings emphasize the vulnerability of plateau regions to influences of elevated emissions, leading to significant changes in BC concentration, mixing

states and light absorption across the TP, which needs to be considered in the evaluation of BC radiative effects for the TP

region.

## 1 Introduction

The Tibetan Plateau (TP) is the largest plateau of the world, covering approximately 2.5 million $km^2$. Its average altitude exceeds 4,000 m and its glaciers cover an area of over 100,000 $km^2$ (Yao et al., 2012a). As the third pole, the TP plays a crucial role in the Asian monsoon systems, the hydrological cycle and global climate (Duan and Wu, 2005; Wu et al., 2007; Wu et al., 2015). Pollutants affect significantly the ecological environment of TP and its surrounding region. They result in increased air temperature (Gustafsson and Ramanathan, 2016), changes in cloud properties (Hua et al., 2020; Lai et al., 2024), glacier retreat (Kang et al., 2010; Kang et al., 2019; Xu et al., 2009; Yao et al., 2012b), anomalies in the hydrological cycle (Luo et al., 2020; Yang et al., 2014; Menon et al., 2002; Ramanathan et al., 2005) and the Asian monsoon (Meehl et al., 2008).

Black carbon (BC) is one of the most important aerosol species affecting climate, glaciers and hydrology in TP (Kopacz et al., 2011; Xu et al., 2009; Yang et al., 2022) because of distinct climate effect (Bond et al., 2013). It is generated by the incomplete combustion of fossil fuels and biomass and is also known as refractory BC (rBC). BC influences the climate directly because it can absorb short-wave radiation (Zhu et al., 2017). The climate forcing of BC is highly dependent on its mixing state. BC can be coated with non-refractory aerosol like organics, nitrate ($NO_3^-$), sulphate ($SO_4^{2-}$) through condensation or coagulation, and turns from externally mixed to internally mixed structure. The mass absorption cross-section (MAC) of BC-containing particles ($PM_{BC}$) can be affected by non-refractory components coated on BC (Cai et al., 2022; Cheng et al., 2016; Gao et al., 2021; Liu et al., 2017; Schnaiter et al., 2005; Wang et al., 2023) via the "lensing effect" (Lack and Cappa, 2010), causing the change in radiative properties of BC. The cloud microphysical properties may also be altered when $PM_{BC}$ are coated with hydrophilic materials and activated into cloud condensation nuclei (CCN), which influences climate indirectly (Bond et al., 2013; Dusek et al., 2006; Henning et al., 2010).

Previous studies have shown that BC has a remarkable direct radiative effect in TP (Zhao et al., 2017; Liu et al., 2021). The radiative effects of BC are not only influenced by its concentration but also by its mixing state. In recent years, there has been an increasing number of field measurements of BC in TP. It is reported that BC concentration can still reach high level occasionally in TP under certain meteorological and synoptic condition (Babu et al., 2011; Zhu et al., 2016; Zhao et al., 2017). Observations on BC mixing states demonstrated that BC is mainly internally mixed (Yuan et al., 2019), and the BC coating enhances the MAC of BC in TP (Wang et al., 2017; Wang et al., 2018; Chen et al., 2019; Tan et al., 2021). BC can be transported over long distance with wildfire plumes (Huang et al., 2023; Zheng et al., 2020). Some regions of TP may be affected by biomass burning (BB) from lower-altitude area (Cao et al., 2010; Zhang et al., 2015; Cong et al., 2015). External transport can raise BC concentration and affect its morphology and mixing state in TP (Tan et al., 2021; Chen et al., 2023). However, research on how emissions from various sources affect the chemical composition of $PM_{BC}$ in TP is scarce. Therefore, we conducted field observations of the physicochemical characteristics of $PM_{BC}$ at two typical sites in TP. The objective of this study is to investigate the impacts of various pollutant emissions and the subsequent regional transport, particularly those from anthropogenic activities from low-altitude regions, on the mixing state and chemical composition of $PM_{BC}$ in TP.

## 2. Materials and Methods

### 2.1 Site Description

Field measurements were conducted at two observation stations in TP (Fig. 1). The station of northeast TP is located in Xihai town (~ 3100 m a.s.l, 36°56' N, 100°54' E). The station of southeast TP is the South-East Tibetan plateau Station for integrated observation and research of alpine environment, located in Lulang (~3200 m a.s.l, 29°46' N, 94°44' E). The field campaign was conducted from April 2 to May 16, 2021 in Lulang and from June 3 to June 23, 2021 in Xihai. Both stations are typical high-altitude sites of mountainous areas (Fig. 1a) but potentially influenced by distinct emission sources. There is more wildfire around Lulang (Fig. 1a), but Xihai is close to the northwest region of China which may largely affected by the anthropogenic emissions (Fig. 1b).

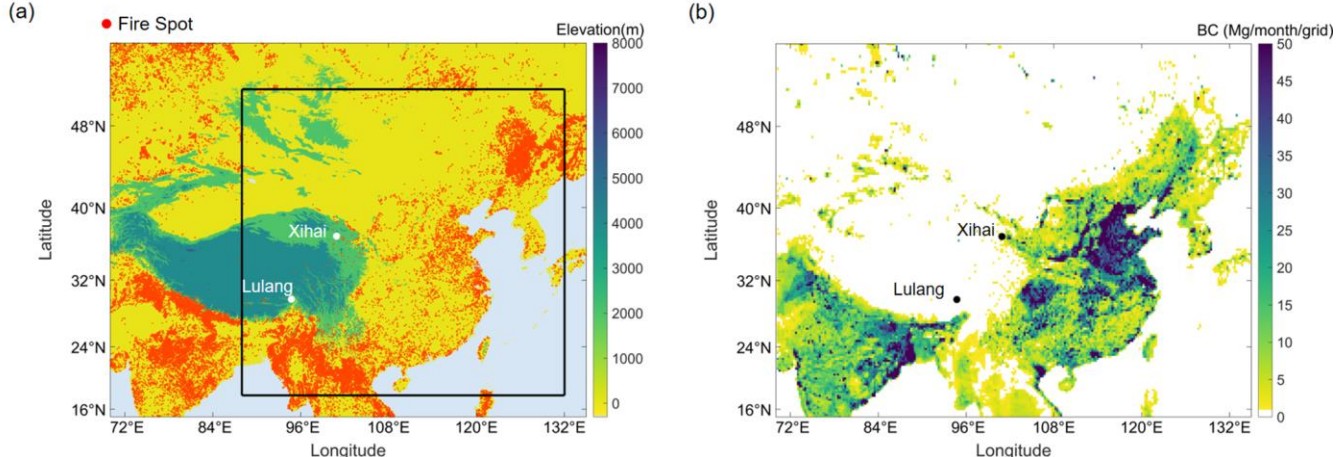

**Figure 1: The maps showing the (a) topographic height and (b) the anthropogenic emissions of BC in the two measurement sites (Xihai, Lulang) and the surrounding region. The red spots represent the wild fire spots during the field measurement period, and the black-line square represents the simulated domain.**

### 2.2 Instrumentation

The Soot Particle Aerosol Mass Spectrometer (SP-AMS, Aerodyne Inc., USA) was used to measure rBC and non-refractory materials coated on rBC (NR-PM$_{BC}$) (Onasch et al., 2012). The tungsten vaporizer was removed and the intracavity infrared laser vaporizer was reserved to exclusively measure PM$_{BC}$. After adjusting the SP-AMS to the laser-only configuration, only PM$_{BC}$ can be volatilized via absorbing laser. We collected V-mode data due to its high sensitivity (Decarlo et al., 2006). The total flow rate through the inlet was maintained at ~3L min$^{-1}$. A PM$_{2.5}$ cyclone was used in the front of the inlet (URG Corp., USA), and only particles in the size range of 50-1000 nm can be focused by the lens of inlet system. The bounce effect of aerosol was eliminated because the tungsten vaporizer was removed, so the usual collection efficiency (CE) (Docherty et al., 2013; Drewnick et al., 2005) is not applicable. The overlap of particle beam and laser beam determined the CE of SP-AMS with laser-only configuration (Willis et al., 2014). The new CE was acquired by intercomparison of rBC concentration measured using SP2 and SP-AMS (Massoli et al., 2015), and was nearly 1 during this campaign.

SP-AMS data was processed by the standard Time-of-Flight AMS data analysis software packages (SQUIRREL version
v1.60P and PIKA v1.20P). Ionization efficiency (IE) calibration was done shortly before removing the tungsten vaporizer. The
mass-based calibration method was used to obtain IE values by sampling the 300 nm dried pure ammonium nitrate particles
into SP-AMS. The 300 nm particles were selected with a differential mobility analyzer (DMA, model 3081, TSI Inc., USA).
The relative IE (RIE) for organic aerosol (OA) and $SO_4^{2-}$ was 1.4 and 1.2, which was consistent to the RIE reported in a
previous work (Canagaratna et al., 2007). The RIE for rBC was calibrated by sampling monodispersed 300 nm Regal Black
particles into SP-AMS. The detection limit was calculated based on the method in Decarlo et al (2006), and the detection limit
of ammonium was higher, so the concentration of ammonium was estimated by ionic equilibrium. OA measured by the SP-
AMS were subdivided into factors with different characteristics and sources based on positive matrix factorization (PMF)
results (Zhang et al., 2005b; Zhang et al., 2011). The PMF Evaluation Tool version 3.04A was used to perform PMF analysis
on the high-resolution organic mass spectra (Ulbrich et al., 2009). Only ions with charge-to-mass ratio below approximately
115 were considered in the PMF analysis.
The meteorological parameters, aerosol optical properties and gaseous pollutants were also measured simultaneously.
Ozone ($O_3$), carbon monoxide (CO), nitric oxide (NO), nitrogen oxides ($NO_x$) and sulfur dioxide ($SO_2$) were measured using
online analyzers (Teledyne API Inc., USA). The photoacoustic extinctiometer (PAX, Droplet Measurement Technologies Inc.,
USA) measured light absorption coefficients. Temperature, relative humidity (RH) and other meteorological parameters were
monitored by meteorological sensors (WXT530, Vaisala Inc., Finland).

**2.3 Model configuration**

In this study, we conducted regional chemical transport modeling using the Weather Research and Forecasting model
coupled with Chemistry (WRF-Chem, version 3.7.1). This model encompasses a broad spectrum of physical and chemical
processes, addressing the emission and deposition of pollutants, advection, diffusion, gaseous and aqueous chemical
transformations, as well as aerosol chemistry and dynamics (Grell et al., 2005). The model domain was centered at 35ºN and
110ºE with a grid resolution of 20 km, covering the northeastern Tibetan Plateau. The vertical structure of the model comprised
30 layers extending from the surface to the top pressure of 50 hPa. The simulation was conducted for the longer period
including the times of whole campaign from 3 June to 11 June 2021.  To establish accurate initial and boundary conditions for
meteorological fields, we updated the model using 6-hourly 1º×1º National Centers for Environmental Prediction (NCEP)
global final analysis (FNL) data. In our pursuit of well capturing the meteorological fields, we assimilated National Centers
for Environmental Prediction (NCEP) Automated Data Processing (ADP) operation global surface observation and global
upper air observational weather data. This assimilation process utilized default nudging coefficients for wind, temperature, and
moisture.
The Yonsei University planetary boundary layer (YSU PBL) scheme was used to parameterize boundary layer processes
(Hong et al., 2006). Other essential physical parameterization options included the unified Noah land surface model (Ek et al.,
2003), the Lin microphysics scheme (Lin et al., 1983), and the Grell-Freitas cumulus parameterization scheme (Grell and
Freitas, 2014). For representing atmospheric chemistry numerically, we utilized the Carbon-Bond Mechanism version Z
photochemical mechanism along with the Model for Simulating Aerosol Interactions and Chemistry aerosol module (Zaveri
and Peters, 1999; Zaveri et al., 2008). Both natural and anthropogenic emissions were considered in this regional WRF-Chem
modeling study. Anthropogenic emissions were derived from the Multi-resolution Emission Inventory for China (MEIC),
which includes emissions from power plants, residential combustion, industrial processes, on-road mobile sources, and
agricultural activities (Li et al., 2017a). Biogenic emissions were calculated online using the Model of Emissions of Gases and
Aerosols from Nature (MEGAN), encompassing more than 20 biogenic species (Guenther et al., 2006).
A comprehensive overview of the model configuration can be referenced in earlier investigations (Huang et al., 2016;
Huang et al., 2018). Additionally, key configurations and validation for the WRF-Chem regional modeling are shown by Table
S1 and Fig. S1.
**2.4 Other materials**
The transport and emission condition were considered to investigate their impacts on BC physical and chemical properties.
The Hybrid Single-Particulate Lagrangian Integrated Trajectory (HYSPLIT) model was used to calculate and cluster 72 h
backward trajectories (Stein et al., 2015; Xu et al., 2018). The starting points of the simulation were Xihai and Lulang, and
particles were released at a height of 1000 m above the ground level. The backward trajectories were calculated every hour
during the field campaign. The Fire Inventory from NCAR (FINN) was adopted to estimate daily open BB emissions with
high spatial resolution (1 km) during the campaign (Wiedinmyer et al., 2006; Wiedinmyer et al., 2011; Wiedinmyer et al.,
2023), and the anthropogenic emissions of major pollutants was estimated by MIX-Asia emission inventory (Li et al., 2017b).
Besides, the optical properties of $PM_{BC}$ were investigated based on the widely-used core-shell Mie model (Bohren and
Huffman. 1983; Virkkula, 2021). MAC and $E_{abs}$ of $PM_{BC}$ were calculated following the algorithm developed by Mätzler (2002).
The refractive index was 1.95 - 0.79i for rBC (Bond and Bergstrom, 2006), and was 1.52 - $10^{-6}$i for BC coating (Pitchford et
al., 2007) at 550 nm wavelength. The calculated optical properties of $PM_{BC}$ in $PM_1$ were validated by good agreements to
observed results of BC in $PM_{2.5}$ (Fig. S2).

## 3. Results and discussion

### 3.1 Overview of BC properties and meteorological conditions in TP

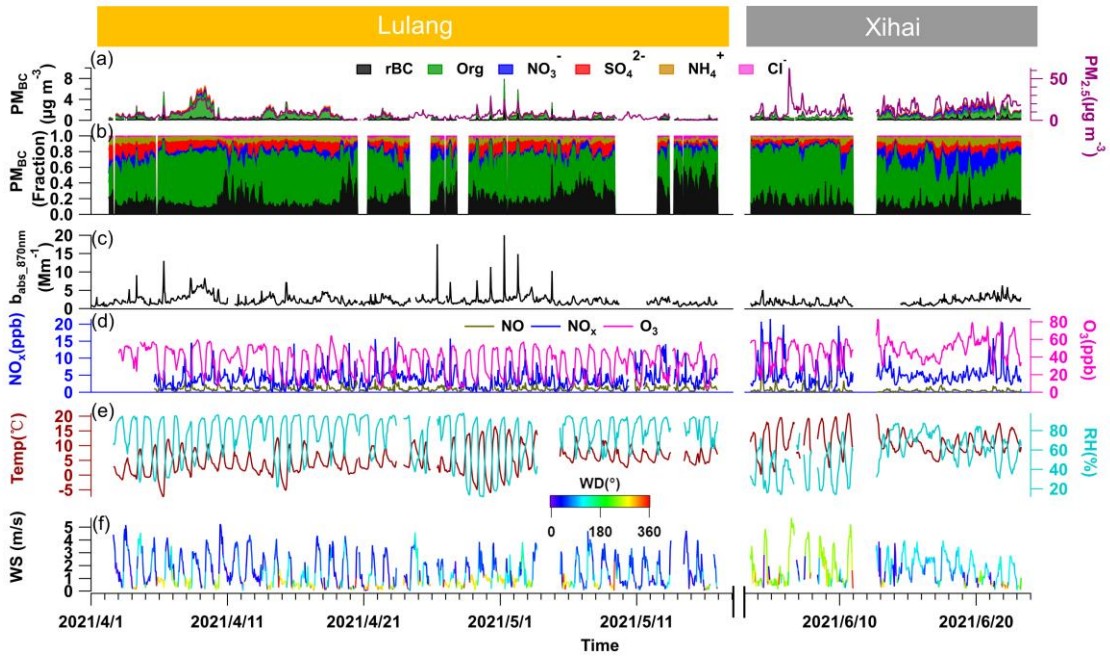

**Figure 2:** The time series of (a) mass concentrations of particulate matters (PM$_{2.5}$), refractory black carbon (rBC), organics (Org), nitrate (NO$_3^-$), sulphate (SO$_4^{2-}$), ammonium (NH$_4^+$) and chloride (Cl$^-$) in PM$_{BC}$, (b) mass fraction of different species in PM$_{BC}$, (c) aerosol light absorption coefficients (b$_{abs}$) at 870 nm wavelength, (d) gaseous pollutants including nitric oxide (NO), nitrogen oxide (NO$_2$) and ozone (O$_3$), (e) air temperature (Temp) and relative humidity (RH), (f) wind direction (WD) and wind speed (WS).

Fig. 2 presents the overall condition during the campaign. The mass concentration of rBC shows large temporal variation at both sites, with ranges of 0.02–1.28 µg m$^{-3}$ in Xihai and 0.02-2.22 µg m$^{-3}$ in Lulang. PM$_{BC}$ concentration and light absorption coefficients (b$_{abs}$) increased in the latter period of Xihai campaign, contrasting with the marked decreasing pattern in PM$_{BC}$ concentration and b$_{abs}$ observed during the latter period of Lulang campaign. In Xihai, the concentration and proportion of inorganic components, especially NO$_3^-$, rose in the latter phase of the campaign as the wind direction (WD) shifted to south-easterly (Fig. 2f). The RH also got higher with the change of wind direction. Another major feature is that the wind direction had distinct diurnal variations. In Xihai, the wind direction converted from easterly and northeasterly flows during the nocturnal hours to southerly direction during daytime. Conversely, Lulang is predominantly controlled by northerly to northeasterly winds throughout the campaign period. Nevertheless, the wind speed (WS) were similar in Xihai and Lulang, with mean value of 1.8±1.2 m s$^{-1}$ and 1.5±1.2 m s$^{-1}$, respectively. In terms of gaseous pollutants, higher levels of NO$_x$ and O$_3$ were observed in Xihai (5.3±3.4 and 48±13 ppb) than in Lulang (4.0±2.5 and 35±15 ppb).

**Table 1: Overview of the BC concentration (mean±1σ) at different sites of TP in existing studies. The minimum value and maximum value were shown in the parenthesis. The measurement result was divided by black lines in the table based on different measurement techniques.**

| Sampling Site | Location | Instrument | Sampling period (Year.Month) | Altitude(m) | BC concentration ($\mu g\ m^{-3}$) | Reference |
|---|---|---|---|---|---|---|
| Lulang | Southeastern TP | SP-AMS | 2021.04-2021.05 | 3300 | 0.17±0.17 (0.02-2.22) | This study |
| Xihai | Northeastern TP | SP-AMS | 2021.06 | 3300 | 0.24±0.20 (0.02-1.28) | This study |
| Qinghai Lake | Northeastern TP | SP2 | 2011.10 | 3200 | 0.36±0.27 (0.05-1.56) | Wang et al., 2014 |
| Nam Co | Central TP | SP-AMS | 2015.05-2015.06 | 4730 | 0.12±0.085 | Wang et al., 2017 |
| Linzhi | Southeastern TP | AE 16 | 2008.11-2009.01 | 3300 | 0.75 (0.30-1.60) | Cao et al., 2010 |
| Lulang | Southeastern TP | AE 16 | 2008.07-2009.08 | 3300 | 0.50±0.52 (0.06-5.37) | Zhao et al., 2017 |
| Mt. Muztagh Ata | Western TP | AE 16 | 2009.11-2010.09 | 4500 | 0.13±0.06 (0.03-0.33) | Zhu et al., 2016 |
| Hanle valley | Southern TP | AE 31 | 2009.08-2010.07 | 4250 | 0.077±0.064 (0.007-0.30) | Babu et al., 2011 |
| Lulang | Southeastern TP | OC/EC Analyzer | 2008.07-2009.07 | 3300 | 0.52±0.35 | Zhao et al., 2013 |
| QOMS | Southern TP | OC/EC Analyzer | 2009.08-2010.07 | 4276 | 0.25±0.22 | Cong et al., 2015 |
| Manora Peak | Southern TP | OC/EC Analyzer | 2005.02-2007.06 | 1950 | 1.0±0.7 (0.1-2.7) | Ram and Sarin, 2009 |

We also compared the observed BC concentration at different sites of TP. Note that, the term "black carbon (BC)" has not been used rigorously or consistently throughout all previous modelling and measurement literature (Bond et al., 2013). Similar terms including "rBC", "equivalent BC (eBC)", and "elemental carbon (EC)" have also been widely used corresponding to different measurement techniques. BC measured by laser-induced techniques is often referred as "rBC", and measured BC using light absorption (e.g. Aethalometer, AE) and thermal/optical methods are normally named as the "eBC" and "EC", respectively. In Table 1, BC concentrations in TP measured by several common techniques were collected and grouped according to the methods to make clearer comparison. Compared to measurements using the same instrument in a metropolitan area (Cui et al., 2022), the rBC concentration of TP ($0.24\pm0.20\ \mu g\ m^{-3}$) was approximately 25% of Shanghai ($0.92\pm0.81\ \mu g\ m^{-3}$). The rBC concentration in Xihai was relatively high compared to southeastern and central TP measured using same technique (Table 1). This was potentially attributed to the strong BC emissions in surrounding area of northeast TP (Fig. 1). The rBC concentration in Lulang exhibited a relatively lower mean value yet with a broad range of variation, suggesting that BC may be subject to diverse airmasses with significant discrepancies in emission intensity across the southeast and southern regions of the TP (Fig. 1). Higher BC levels were observed at stations in proximity to the Indo-China Peninsula and South Asia where wildfire activities were extremely intense in spring. Therefore, the considerable variability of rBC concentrations in Lulang is likely due to the alternating influences from airmasses transporting BB plume and those originating from cleaner environments.

**3.2 Physicochemical characteristics of BC-containing particles in TP**

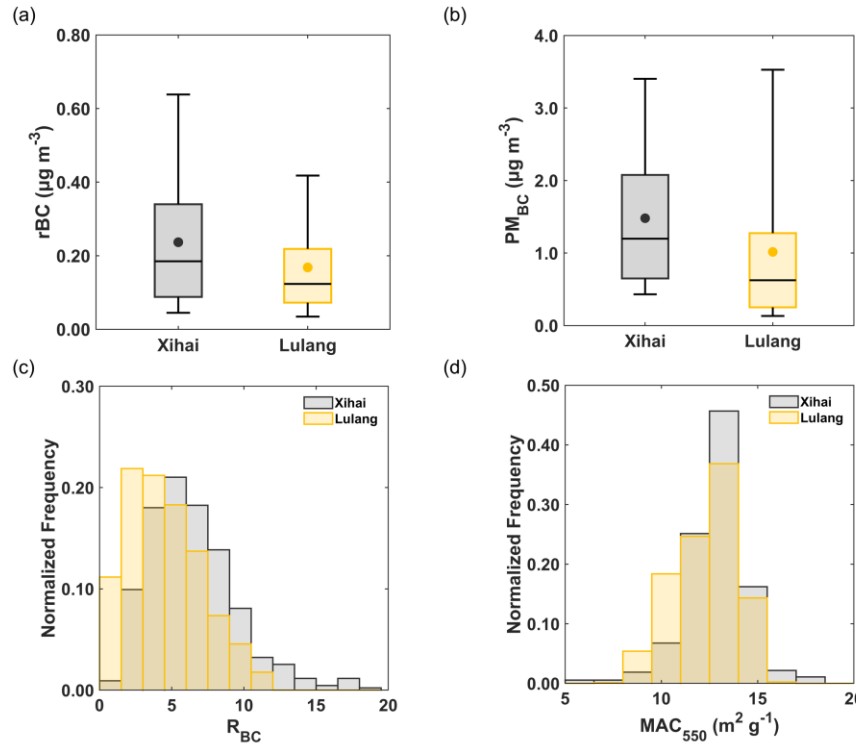

**Figure 3: The box plots of (a) rBC and (b) BC-containing particles mass concentrations in Xihai and Lulang, the lower and upper lines of box plot represent the 25th and 75th percentiles and the whiskers stand for 5th and 95th values. The charts of normalized frequency distribution show (c) mass ratio of coating substance to rBC core ($R_{BC}$) and (d) mass absorption cross-section (MAC). Only 1.15% of the $R_{BC}$ exceeded the maximum value of bin (19.5) in Xihai, and no $R_{BC}$ exceeded the maximum value of bin in Lulang.**

The overall characteristic of $PM_{BC}$ in Xihai and Lulang was compared based on statistical results. As Fig. 3a and Fig. 3b show, the mass concentration of rBC and $PM_{BC}$ were higher in Xihai due to possible impacts of stronger anthropogenic emissions (Fig. 1b), and the difference ($t_{rBC}$=2.8, $t_{PMBC}$=2.1) between the two sites was proved by the t-test ($\alpha$=0.05, $v$=50). Figure 3c compares mixing state of $PM_{BC}$ in Xihai and Lulang, which was expressed by the mass ratio of BC coating to rBC ($R_{BC}$). The frequency distribution of $R_{BC}$ had obvious difference at two sites. $R_{BC}$ in Xihai was generally higher than in Lulang, indicating the thicker coating in Xihai. The peak of $R_{BC}$ occurred at [4.5,6] and [1.5,3] in Xihai and Lulang, respectively. $R_{BC}$ of more than 50% $PM_{BC}$ was between 3.0 and 7.5, and only 11% $PM_{BC}$ had $R_{BC}$ less than 3.0 in Xihai. Unlike Xihai, the percentage of thinly coated $PM_{BC}$ that $R_{BC}$ was less than 3.0 was higher to 33% in Lulang. The difference on mixing states of $PM_{BC}$ was also demonstrated by the t-test ($t_{RBC}$=2.4). The peak of MAC at both sites was between 12 and 14 $m^2$ $g^{-1}$ (Fig. 3d) which was significantly greater than the MAC of BC without coating (Bond and Bergstrom, 2006), and average value and range of MAC in Xihai and Lulang was 12.8 (5.6-17.4) and 12.3 (6.8-15.7) $m^2$ $g^{-1}$. Over 61% of BC was distributed in larger MAC range (higher than 12.5 $m^2$ $g^{-1}$) in Xihai, showing stronger light absorption ability of BC in this region. Due to the synergy of higher mass concentration and light absorption ability, $PM_{BC}$ could bring larger climate effects in northeast TP.

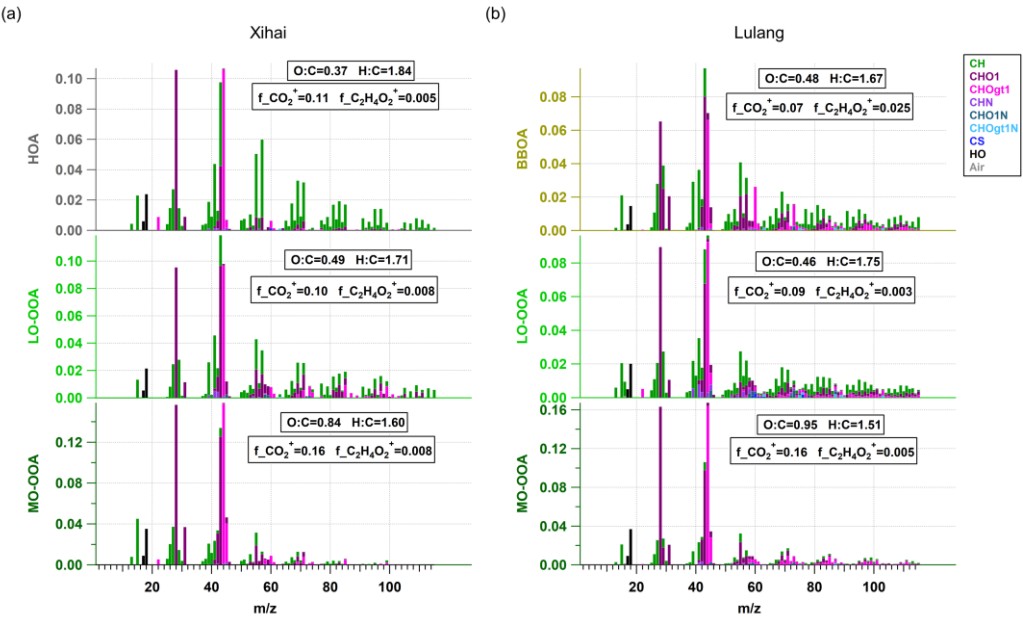

**Figure 4: The mass spectra of different factors represents the organic aerosol from specific sources in BC-containing particles in (a) Xihai and (b) Lulang. MO-OOA is more oxidized oxygenated organic aerosol, LO-OOA is less oxidized oxygenated organic aerosol, HOA is hydrocarbon-like organic aerosol and BBOA is biomass burning organic aerosol.**

The chemical characteristics and sources of OA in $PM_{BC}$ were identified by PMF. OA was separated into primary OA (POA) and oxygenated OA (OOA) at both sites (Fig.4 and Fig. S3). In Xihai, there were one factor originating from primary emissions and two factors from secondary formation. The POA factor had higher signal of $C_4H_7^+$ and $C_4H_9^+$, which is the important alkyl fragments from primary sources (Hu et al., 2016), in its mass spectrum. It also had higher content of hydrogen that H:C was up to 1.84 and lower signal of $C_2H_3O^+$ which is the typical BB tracer. Hence, this factor was mainly emitted from fossil fuel combustion rather than BB, and was named as Hydrocarbon OA (HOA). OOA factors were further divided into less−oxidized OOA (LO-OOA) and more−oxidized OOA (MO-OOA) factors. These two factors were secondary OA (SOA) formed through oxidation processes such as photochemical reactions (Kanakidou et al., 2005; Zhang et al., 2005a; Zhao et al., 2018). They had higher fraction of signal of $CO_2^+$ ion (m/z 44) and other oxygenic ions in mass spectrum, which is similar to the mass spectra of typical OOA reported in other field campaigns (Crippa et al., 2013; Hu et al., 2016; Kim et al., 2020; Lee et al., 2017; Sun et al., 2016; Sun et al., 2020; Wang et al., 2016; Zhou et al., 2018). The O:C of the two OOA factors was also calculated (Canagaratna et al., 2015) to learn about the oxidation degree of OOA. MO-OOA exhibited higher O:C ratio (0.84) than LO-OOA (0.49). Unlike Xihai, the POA factor in Lulang had higher fraction of signal of $C_2H_3O^+$ (m/z 60) ion ($f\_C_2H_3O^+$) in mass spectrum, which is the fragment of levoglucosan mainly from BB (Lee et al., 2010). Therefore, this POA factor was identified as biomass burning OA (BBOA) in Lulang. Moreover, the $f\_CO_2^+$ and $f\_C_2H_4O_2^+$ (0.065 versus 0.025) of this factor were also within the triangle area in previous BBOA study (Cubison et al., 2011), and the $f\_C_2H_4O_2^+$ was lower than the fresh BBOA, indicating that this factor was influenced by biomass burning activities and aging processes collectively. The remaining

two factors were from SOA formation in Lulang, and had higher fraction of signal of $CO_2^+$ ion. Based on the oxidation degree,

the two factors were identified as MO-OOA and LO-OOA. The O:C of MO-OOA and LO-OOA was 0.95 and 0.46,

respectively. Compared to Lulang, the OA in BC coating was under stronger impacts of anthropogenic emissions in Xihai

indicated by HOA.

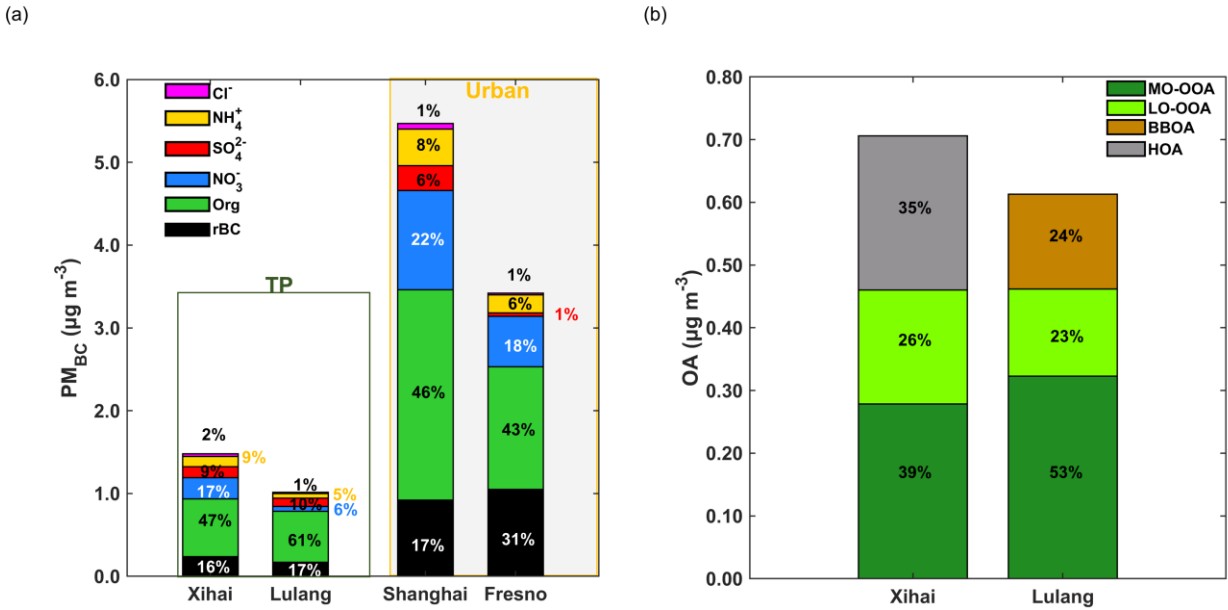

**Figure 5: The stacked bars represent mass concentrations of (a) different species in BC-containing particles (PM$_{BC}$), and (b) different factors of organic aerosol in BC-containing particles. The numbers on the plot show the percentage of different species and organic factors. In subplot (a), PM$_{BC}$ in the TP (this study) was compared to PM$_{BC}$ in urban regions (Collier et al., 2018; Cui et al., 2022).**

Figure 5 presents PM$_{BC}$ chemical composition at two sites. BC coating had higher mass contribution to PM$_{BC}$ in Xihai

and Lulang compared to the urban site (Collier et al., 2018), indicating the thick coating of PM$_{BC}$ in TP. The average mass
fraction and concentration of BC coating were 84% and 1.2 μg m$^{-3}$ in Xihai. The mass fraction of coating was similar (83%)
in Lulang, although the concentration of BC coating was lower (0.85 μg m$^{-3}$). OA was the dominant component of BC coating
(Fig. 5a) at both sites, which was consistent with the observation in central TP (Wang et al., 2017). OA took up a higher
proportion in BC coating in Lulang compared to Xihai, Shanghai (Cui et al., 2022) and Fresno (Collier et al., 2018). During
the field campaign, the average concentration of HOA, LO-OOA and MO-OOA was 0.25, 0.18 and 0.28 μg m$^{-3}$ in Xihai. MO-
OOA also had the highest concentration (0.32 μg m$^{-3}$) of OA in Lulang, and exceeded BBOA (0.15 μg m$^{-3}$) and LO-OOA
concentration (0.14 μg m$^{-3}$). It demonstrated that SOA formation plays an important role in coating process of PM$_{BC}$. The BC
coating was dominated by MO-OOA which was importantly affected by atmospheric oxidizing process. The concentration of
$O_3$ highly relative to atmospheric oxidizing capacity improved significantly in afternoon (Fig. S8), and the enhanced oxidizing
capacity could cause increase of MO-OOA in BC coating in both Xihai and Lulang. Besides MO-OOA, $NO_3^-$ (17%) and HOA
(35%) also made large contribution on BC coating (Fig. 5a) and coated OA (Fig. 5b) in Xihai compared to Lulang. The HOA
and $NO_3^-$ were both closely associated with anthropogenic sources because the anthropogenic sources emitted the HOA (Zhang
et al., 2005a) and precursors of $NO_3^-$ largely (Dall'osto et al., 2009; Richter et al., 2005; Sun et al., 2018). It indicated that
anthropogenic emissions have a strong influence on coating process of $PM_{BC}$ in northeast TP, which is quite different from
southeast TP.

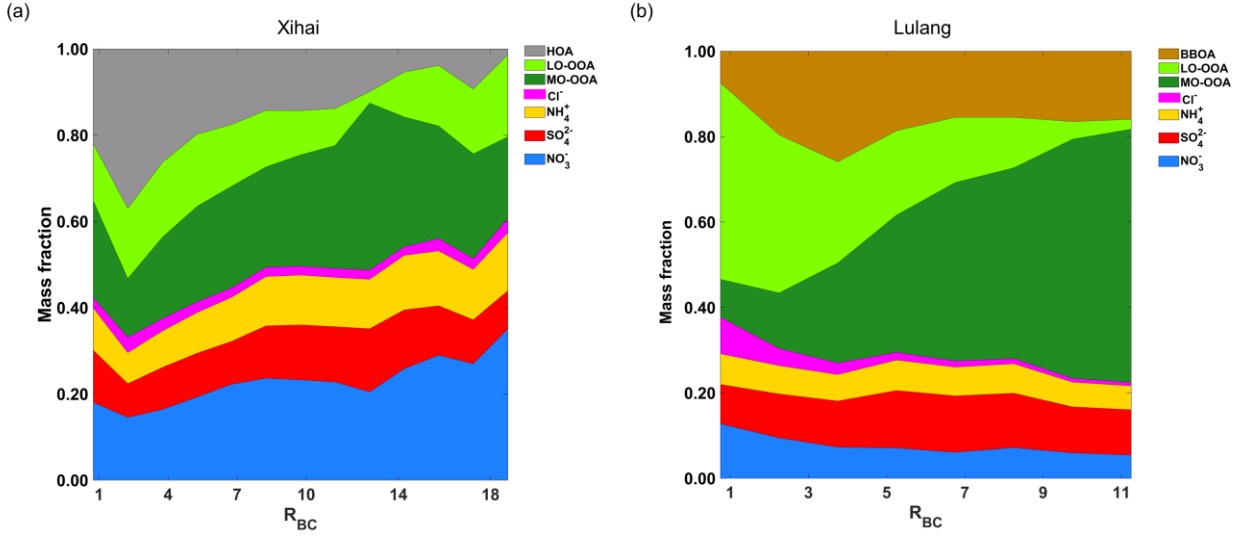


**Figure 6: The variation of BC coating composition with $R_{BC}$ between (a) Xihai and (b) Lulang. The x-axis represents the mass ratio**
**of BC coating components and rBC cores ($R_{BC}$), and the y-axis represents the mass fractions of BC coating components coated on**
**rBC. The mass fraction of components was averaged in each bin of $R_{BC}$ (bin width: 1.5).**
Figure 6 shows the coating components of BC with different $R_{BC}$ in Xihai and Lulang. The mass fraction of MO-OOA
was predominant in the thick-coated $PM_{BC}$ in both Xihai and Lulang. Notably, a more significant enhancement in MO-OOA
contribution within the thickly coated $PM_{BC}$ was exhibited in Lulang, concomitant with a reduced fraction of inorganic
components. The mass fraction of MO-OOA was only 9% in the thin BC coating ($R_{BC}<1.5$), rising dramatically to 59% in
those with $R_{BC}$ exceeding 10.5 (thick BC coating). Another notable feature of the coating components was the higher
contribution of BBOA in Lulang, especially when the coating thickness of $PM_{BC}$ was higher. It indicated that thickly coating
of BC was affected by BB activities and atmospheric oxidation significantly. In contrast to Lulang, HOA contribution
decreased with the growth of $R_{BC}$, indicating a weaker effect of primary aerosol on thickly-coated $PM_{BC}$ in Xihai. Besides the
MO-OOA, $NO_3^-$ also contributed significantly to the composition of thickly-coated $PM_{BC}$ in Xihai, while the contribution of

NO$_3^-$ dropped with the rise of R$_{BC}$ in Lulang. As illustrated in Fig. 6a, the mass fraction of NO$_3^-$ reached to 35% in the maximum bin of R$_{BC}$ (18-19.5) in Xihai. The abundant NO$_3^-$ was closely associated with anthropogenic sources as mentioned in the preceding paragraph. The results demonstrate substantial variability in the composition influencing BC aging across TP affected by diverse emission sources. Moreover, anthropogenic pollutant emissions had strong impacts on BC coating even in the remote highland areas, and the contribution of inorganic aerosol to BC coating is non-negligible in TP.

## 3.3 Impacts of transported emissions on BC-containing particles

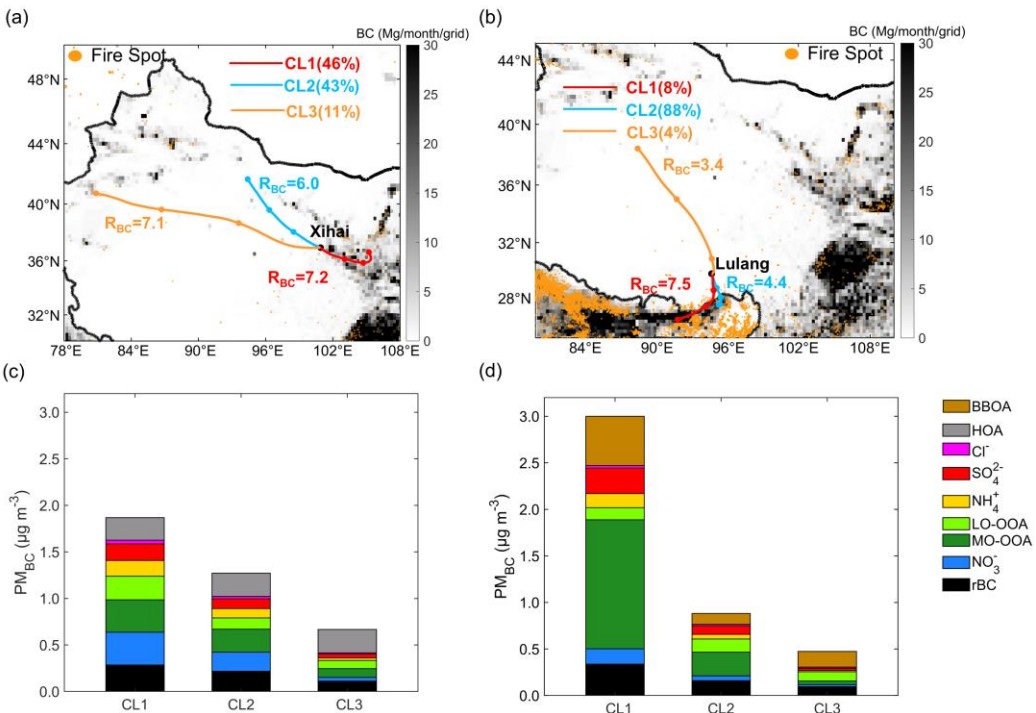

**Figure 7: The maps show the backward trajectories in different clusters of (a) Xihai and (b) Lulang. Each circular marker along the trajectories denotes a 24-hour interval. The background shading represents the anthropogenic BC emission intensity and the orange spots represent the location of wildfire during the campaign in (a) and (b). The stacked bar plots show the mass concentration of coating components and rBC in (c) Xihai and (d) Lulang.**

As discussed above, PM$_{BC}$ in TP region is possibly affected by both anthropogenic sources and BB transported from surrounding areas. To further investigate the impact mechanism of regional transport on BC, the cluster analysis of backward trajectories was carried out during field campaign of Xihai and Lulang, and backward trajectories were clustered into three kinds. In Xihai, the airmasses were dominantly from eastern region outside of TP, as indicated by airmasses cluster1 (CL1), followed by the airmasses of cluster2 (CL2) from the northwest of Xihai, and the airmasses of cluster3 (CL3) from west of Xihai (Fig. 7a). PM$_{BC}$ was brought more to Xihai (Fig. 7c) by the airmasses of CL1 which went through the lower-altitude regions with stronger anthropogenic BC emissions (Fig. 7a and Fig. 1b). In Lulang, the CL1 airmasses from South Asia were

heavily polluted and aged, the CL2 airmasses from southern edge of Himalayas and the CL3 airmasses from central inland of
TP were cleaner (Fig. 7b). Comparing the polluted airmasses (CL1) at two sites, chemical composition of $PM_{BC}$ showed
obvious difference between Xihai and Lulang (Fig. 7c and 7d). The contribution of inorganic species to BC coating was higher
in Xihai, and there was more OA (especially MO-OOA) in polluted airmass of Lulang. MO-OOA was the major component
of BC coating in CL1 in Lulang. As shown by Fig. 7b, there was intensive wildfire in the source region of CL1 airmasses of
Lulang, and the wildfire plume could be readily uplifted to higher altitude due to prevailing upflow driven by the lifting of the
plume (Freitas et al., 2007; Fromm et al., 2000; Labonne et al., 2007; Luderer et al., 2006; Sofiev et al., 2012) or large-scale
westerly and small-scale southerly circulations during the pre-monsoon season (Zhang et al., 2020; Cao et al., 2010). Such
circulation could transport BC and other co-emitted pollutants from wildfires in Indo-China Peninsula and South Asia over the
mountain of TP and reached Lulang. Because the biomass burning during wildfires can emit plentiful volatile organic
compounds (VOCs) like terpenes (Akagi et al., 2013; Fiddler et al., 2024), it is expected that SOA can be formed through
oxidation from precursors in the plume, leading to a thick coating on $PM_{BC}$. In Xihai, $NO_3^-$ was one of the major coating
species in $PM_{BC}$ in CL1 (Fig. 7c) with mass concentration of $NO_3^-$ up to 0.35 μg m$^{-3}$ (accounts for 19% of $PM_{BC}$), and other
airmasses clusters had higher mass fraction of HOA in BC coating indicating that $PM_{BC}$ was less affected by oxidation and
was fresher. CL1 transported from northwest region of China where the anthropogenic emissions are much stronger than TP
(Fig. 7a). With higher concentrations of primary pollutants like $NO_x$, the formation and coating of $NO_3^-$ can be enhanced in
$PM_{BC}$. Above results indicated that the effects of emission sources were discrepant in different regions of TP, and the northeast
part of the TP was significantly affected by anthropogenic emissions.

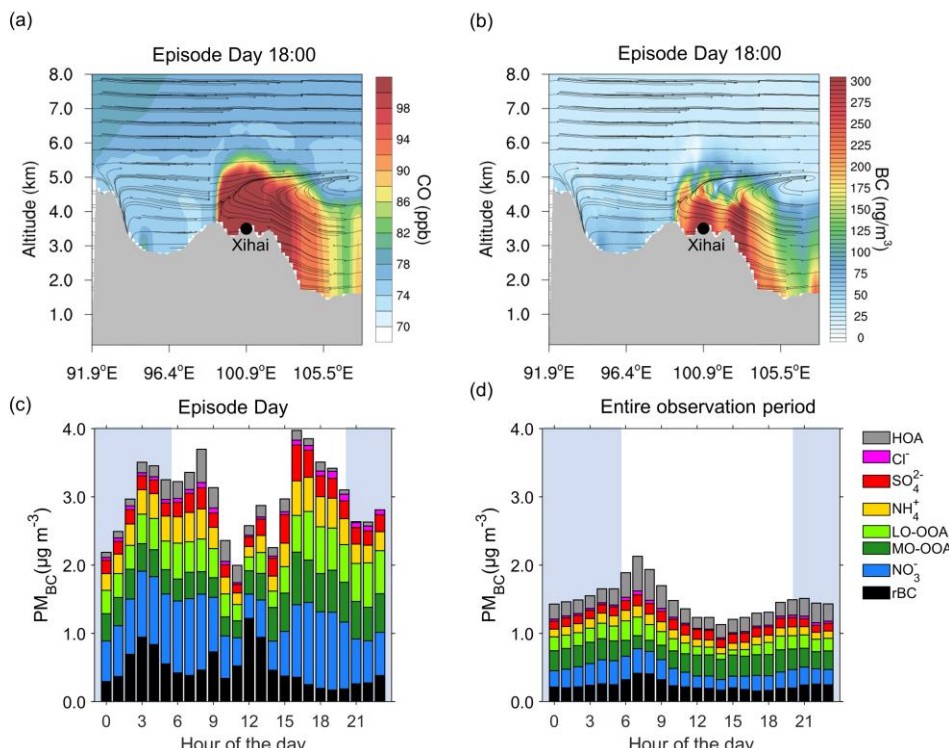


**Figure 8: Simulated meridional mean concentration profile of (a) CO and (b) BC independently during the episode day (19 June, 2021). The air circulation is shown as vector arrows and the terrain height is shown as gray shade in (a) and (b) subplots. The vertical velocity of wind was amplified by a factor of 3000 for clarity. The (c) and (d) subplots show the diurnal variation of BC-containing particles concentration during the (c) episode day and (d) entire observation period in Xihai. The blue shade represents the nighttime hours during Xihai campaign in (c) and (d) subplots. The sunrise on Xihai was about 6:00 a.m. (Beijing Time), and sunset was about 8:30 p.m. (Beijing Time).**

To further explore the coupling effect of horizontal and vertical transport on BC in high-altitude region, both observation and simulation were performed to track the evolution of pollutants in surrounding area. We chose a typical episode in CL1 in Xihai to conduct model simulation. As illustrated in the meridional profile plots of CO and BC, the high levels of anthropogenic pollutants were uplifted to Xihai (Fig. 8a and Fig. 8b). The updraft flow and the turbulent mixing in the boundary layer carried the anthropogenic emissions from the ground to the high altitude, and then the horizontal easterly winds transported the anthropogenic emissions to the northeast TP. The combination of upward wind and developing boundary layer (Fig. S8c) allowed the pollutants emitted by the anthropogenic sources near the surface to be carried aloft and transported to high-altitude TP in the afternoon. This effect can significantly change both the concentration and chemical composition of BC. Compared to the average diurnal variation during observation period, the diurnal variation during episode shows distinctive features (Fig. 8c and 8d). $PM_{BC}$ concentration increased remarkably from 15:00 and peaked at 16:00 to 17:00 with a maximum concentration of 4.0 $\mu g\ m^{-3}$. Concurrently, $NO_3^-$ and SOA also exhibit a noticeable increase along with the thickening BC coating in the afternoon. The $NO_3^-$, SOA, and $R_{BC}$ rose from 0.41 $\mu g\ m^{-3}$, 0.49 $\mu g\ m^{-3}$, and 2.8 at 11:00 to 1.06 $\mu g\ m^{-3}$, 1.31 $\mu g\ m^{-3}$, and 10.2 at 16:00, respectively. As the Fig. S8a shows, $O_3$ did not increased significantly after 3:00 p.m. in Xihai, implying that

the photochemistry and secondary aerosol formation might not enhance. However, the consistent radiative heating of the
ground surface during the daytime kept a convective boundary layer (Fig. S8c), facilitating the vertical transport of
anthropogenic emissions to higher altitudes and plausibly causing the enhanced air pollution in the afternoon in Xihai. This
phenomenon is a good illustration of the vulnerability of remote plateau regions to intense anthropogenic influences, as
pollutants can be transported from low-altitude regions to the plateau.
**3. 4 Impacts of diverse BC coating characteristics on light absorption**

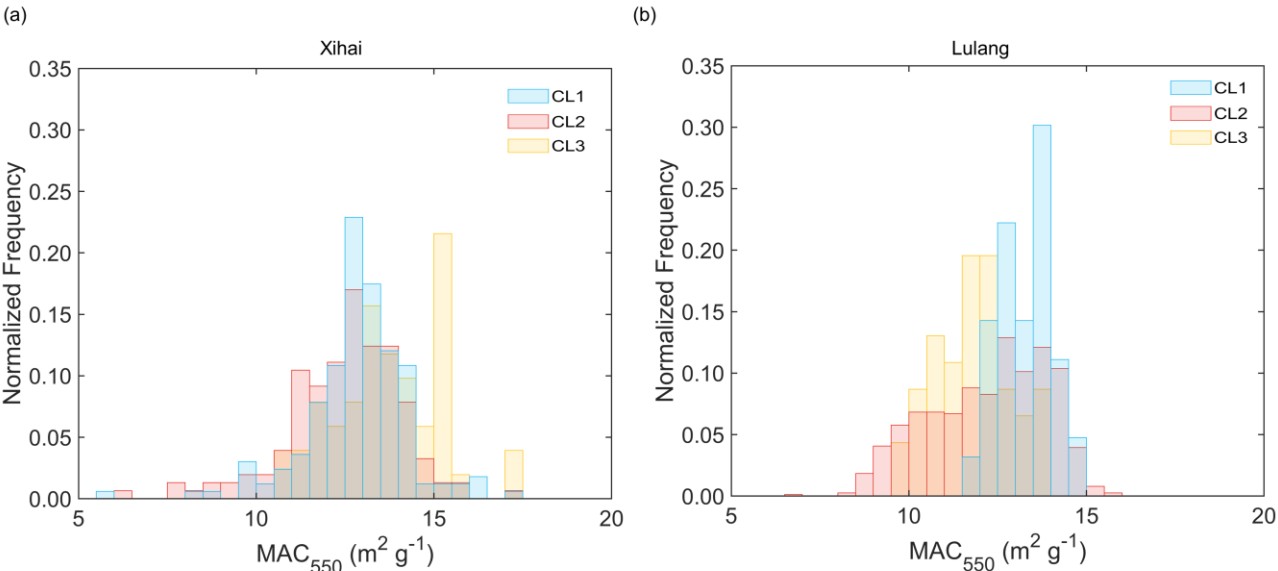


**Figure 9: The normalized frequency distribution of MAC at 550 nm wavelength in different trajectories clusters of (a) Xihai and (b)**
**Lulang.**
The effects of different emission sources on the BC light absorption ability were investigated. Compared to Lulang, the
MAC of $PM_{BC}$ was overall higher in Xihai, indicating higher absorption efficiency and potentially stronger radiative forcing
in this region. The MAC were all relatively high in three clusters of airmasses of Xihai, with distribution peaked between 12
and 14 $m^2\ g^{-1}$ that numerically comparable to previous studies (Wang et al., 2015). The overall high MAC in Xihai may result
from the significant impact of anthropogenic emissions in northeast TP. The stronger emissions provided abundant precursor
of BC coating to improve the coating thickness, and the thick coating enhance light absorption capacity of $PM_{BC}$ via "lensing
effect". While MAC was higher only under control of the polluted CL1 airmasses in Lulang, indicating that the South Asian
wildfire plume could significantly strengthen the light absorption ability of BC. The MAC in Lulang was also comparable to

previous studies (Wang et al., 2018) that the peak of MAC distribution was 7.6 m² g⁻¹ at 870 nm (12.0 m² g⁻¹ at 550 nm if the Absorption Ångström Exponent of BC is 1.0). In CL1 airmasses of Lulang, MAC mainly distributed at the bin between 12 and 14 m² g⁻¹ that is close to MAC (13.1 m² g⁻¹ at 550 nm) at other TP sites affected by biomass burning plume (Tan et al., 2021). The BC coating was thick (Fig. 7d) to improve the MAC in CL1 airmasses of Lulang influenced by higher BB emissions. These results indicate that strong BB and anthropogenic emissions from surrounding area could make noticeable impacts on chemical composition and light absorption ability of BC in TP, and these impacts were more prevalent in the northeast part of the TP.

## 4 Conclusions

In this study, we employed the SP-AMS only with a laser vaporizer to quantitatively analyze the chemical composition of $PM_{BC}$ at distinct sites, Xihai and Lulang, located in the northeast and southeast regions of the TP. Our findings demonstrate the considerable variability and spatial heterogeneity of BC physical and chemical properties across the TP. Notably, Xihai exhibited higher mass concentrations of rBC and $PM_{BC}$, with respective mean concentrations of 0.24 μg m⁻³ and 1.48 μg m⁻³, compared to 0.17 μg m⁻³ and 1.02 μg m⁻³ in Lulang. The $PM_{BC}$ in Xihai has higher aging degree, as indicated by a higher mean $R_{BC}$ of 6.7, contrasting the mean $R_{BC}$ of 4.5 in Lulang.

The marked differences in chemical composition of $PM_{BC}$ were also observed within TP region. Due to differences in emission sources, the POA was distinct in Xihai and Lulang. HOA from fossil fuel combustion was one of the main components of $PM_{BC}$ in Xihai as the result of elevated anthropogenic emissions, and there was more BBOA in Lulang especially when the airmasses were from South Asia Plain affected by frequent wildfire. Besides primary species, the secondary coating components also showed larger differences. The contribution of secondary inorganic aerosols, particularly $NO_3^-$ was noticeably higher in Xihai because of the strong anthropogenic emission of $NO_x$ as the precursor of $NO_3^-$. SOA was comparatively higher in areas with less anthropogenic emissions like Lulang. The oxidizing level of SOA was high in both sites of TP that the MO-OOA occupied the largest mass fraction of SOA. We also investigated the variation of $PM_{BC}$ composition with its coating thickness in both sites. An enhancement in $NO_3^-$ fraction was observed on aged BC coating in Xihai. In contrast, the mass contribution of $NO_3^-$ decreased and SOA contribution notably increased during the thickening of $PM_{BC}$ in Lulang.

Backward trajectory analysis and regional chemical transport modeling were then performed to track the impacts of transported anthropogenic and BB emissions on chemical composition of $PM_{BC}$ in northeastern and southeastern TP. The effect of anthropogenic emissions was stronger in northeastern TP when the airmasses were brought by updrafts and easterly winds from lower-altitude areas, leading to an increase of $NO_3^-$ and SOA coated on BC. With the development of boundary layer, strong turbulent mixing promoted the elevation of anthropogenic pollutants. In contrast to Xihai, the thickly coated BC in Lulang was mainly caused by elevation and transportation of biomass burning plume from the South Asia, leading to a significantly higher contribution of MO-OOA and BBOA. The distinct transported emissions caused substantial variations of

chemical composition and mixing state of BC, which further changes the light absorption ability of BC in the TP. The MAC
of PM$_{BC}$ at both sites was at a high level, showing the strong absorption ability of BC in TP region, especially in polluted
airmasses affected by biomass burning emission from the South Asia. The overall thicker coating and higher MAC of PM$_{BC}$
in airmasses elevated from lower-altitude regions reveals the impacts of promoted BC aging processes during transportation
on the mixing state and light absorption of BC in TP, which will further influence its radiative effects. Such impact needs to
be considered in the evaluation of BC radiative effects for the TP region.

**Data availability**

The wildfire emission data FINN is available at https://www.acom.ucar.edu/Data/fire/. The anthropogenic emission data MIX
is available at http://www.meicmodel.org/dataset-mix.html. The BLH is acquired from the fifth-generation European Centre
for Medium-Range Weather Forecasts (ECMWF) reanalysis data (ERA5; https://cds.climate.copernicus.eu/cdsapp#!/home).
The measurement data covered in the article can be found at: https://doi.org/10.6084/m9.figshare.25399024. Additional data
related to this paper may be requested from the authors.

**Author contribution**

CF, AD, and JPW conceptualized and supervised this study. JBW, YZ, TL, XC, DG, CZ, LW, XQ and WN conducted the
field campaign. JBW and JPW conducted the data analysis. SL and XH contributed to the model development and simulation.
JBW wrote the draft and drew the plots. JPW, XH and QZ discussed the results. JBW and JPW reviewed and edited the paper
with contributions from all co-authors.

**Competing interests**

The contact author has declared that none of the authors has any competing interests.

**Acknowledgments**

This work was supported by the second Tibetan Plateau Scientific Expedition and Research (STEP) program (2019QZKK0106)
and the National Natural Science Foundation of China (42005082).

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
