# Peer review of "Impacts of elevated anthropogenic emissions on physicochemical characteristics of BC-containing particles over the Tibetan Plateau"

_EGUsphere, 2024_

## Author Comment (AC1)

**Response**

The manuscript by Wang et al. provides a comprehensive analysis of the physicochemical properties of black carbon (BC)-containing particles over the Tibetan Plateau, with an emphasis on the impacts of anthropogenic emissions. The authors conduct detailed field observations, which contribute valuable data to this field of study. Their findings represent a noteworthy advancement in elucidating the impact of anthropogenic emissions on the properties of BC, especially within the environmentally sensitive Tibetan Plateau region. See my detailed comments below.

We appreciate the reviewer's kind effort and insightful comments. The amendment and modification have finished followed by all constructive comments in the revised manuscript and supporting information. Please kindly find our point-by-point responses listed below. The reviewer's comments are in blue font followed by our responses and revisions in the manuscript (in Italic).

**Major Comments:**

1. **Addressing Seasonality and Expanding Temporal Scope:** The manuscript effectively outlines how regional transport influences the BC characteristics over the plateau. This is particularly evident in the comparison between the northeast and southeast regions. Yet, the study seems to focus predominantly on the spring season (Table 1). It would be valuable if the authors could discuss the potential seasonality of these findings or provide reasons for the focus on this particular season, including how the results might vary in other seasons.

**Response 1**

Thanks so much for your foresighted suggestions. Due to the limitation of experimental condition, it is very hard to conduct long-term continuous observation at these sites in TP. Therefore, we did intensive observation focusing on the specific targets and we will conduct field campaign in more seasons in the future to explore the seasonal variation of BC characteristics.

The Xihai observation was designed to study the transportation of stronger anthropogenic emission from lower altitude regions in northwestern China. As Fig. 7a shows, Xihai was under relatively strong influence of transported anthropogenic emission from lower altitude regions during this period.

[Figure]

*Figure 7 (a): The maps show the backward trajectories in different clusters of Xihai.*

The Lulang observation was focused on studying the impact of biomass burning (BB) on BC characteristics over Southeastern TP. The average level of BC emissions from wildfires in surrounding area of Lulang was much higher during the observation period (93381 kg d$^{-1}$) than other periods (33877 kg d$^{-1}$), which provide us ideal condition to investigate the impacts of BB on BC physical and chemical properties.

2. **Lack of Detailed Model Evaluation**:  The manuscript utilizes the WRF-Chem model to simulate
the atmospheric processes and black carbon (BC) characteristics over the Tibetan Plateau. However, there
seems to be a lack of detailed evaluation or validation of the model simulations against observational data.
Without proper validation, the reliability of the model results and the subsequent conclusions drawn from
them may be questionable.

**Response 2**

Thanks so much for your suggestions about the validation of model in our manuscript. We have
added the evaluation and validation of model simulations into the revised SI in Line 20-32:

[Figure]

*Figure S1: The time series of the near-surface air temperature, sulfur dioxide ($SO_2$), ozone ($O_3$) and mass concentration of fine*
*particulate matter ($PM_{2.5}$) in the Xihai and surrounding area. The line and marker represent the results of ambient measurement and*
*modelling respectively. The MB and NMB are mean bias and normalized mean bias of each parameter.*

*In this study, the air temperature at 2 m were evaluated based on the measurement data from our*
*measurement and publicly available meteorological datasets of the University of Wyoming (http://www.*
*weather.uwyo.edu/surface/). The air temperature was pretty close between the modelling and*
*measurement, and the mean bias was +1.00 ℃. It was shown that the model had a good performance in*
*the simulation of meteorological fields.*

*The air quality dataset at Xihai and the monitoring stations near to Xihai (https://quotsoft.net/air/)*
*were used to evaluate the WRF-Chem model in simulating the air pollution. There were overall good*
*agreement and small bias between model-simulated and observed concentration values of gaseous*
*pollutant ($SO_2$, $O_3$) and particulate matter. The modelled $SO_2$ concentration level is relatively low, however,*
*it is not the pollutants of major concern in this study.*

**Minor Comments:**

3.Line 155: According to my understanding, the study by Cui et al. (2022) only provides BC
concentrations in urban areas of Shanghai, and does not directly provide the number 25%. Please directly
provide the BC concentrations mentioned in Cui et al. (2022)'s study and explain how the calculation for
obtaining 25% is done. In addition, although EC is sometimes used as a substitute for BC in some cases
due to the lack of BC observations, it should not be said that they are equivalent.

**Response 3**

Thanks so much for your suggestions, and we have modified the relevant description.

Firstly, the 25% ratio was calculated based on Cui et al. (2022) and our study. In Cui et al. (2022), the campaign-average rBC concentration measured by SP-AMS was $0.92 \pm 0.81$ µg m$^{-3}$ in Shanghai which is a typical urbanized area. In our study, the average rBC concentration was $0.17 \pm 0.17$, $0.24 \pm 0.20$ µg m$^{-3}$ in Lulang and Xihai, respectively. Lulang and Xihai represent the general environment in different regions of the Tibetan Plateau.

Since these two studies used the same instrument, the measured concentration level can be compared directly. The ratio of rBC concentration in Xihai and Shanghai was approximately 25%, and for Lulang, the average rBC concentration was 18% to that in Shanghai. We have modified the expression in the manuscript to make the meaning of this ratio clearer:

Line 176-178: *Compared to measurements using the same instrument in a metropolitan area (Cui et al., 2022), the rBC concentration of TP (0.24 $\pm$ 0.20 µg m$^{-3}$) was approximately 25% or less of Shanghai (0.92 $\pm$ 0.81 µg m$^{-3}$).*

Secondly, thanks you for reminding us to add the necessary clarification about the differences in measurement methods for black carbon. "BC (EC)" here is not very precise. The term "black carbon (BC)" has not been used rigorously or consistently throughout all previous modelling and measurement literature. Similar terms including "rBC", "eBC", and "EC" has also been widely used corresponding to different measurement techniques. We have corrected the heading of Table 1 as "BC concentration" and modified the Line 170-176 of manuscript:

*Note that, the term "black carbon (BC)" has not been used rigorously or consistently throughout all previous modelling and measurement literature (Bond et al., 2013). Similar terms including "rBC", "eBC", and "EC" has also been widely used corresponding to different measurement techniques. BC measured by laser-induced techniques is often referred as "rBC", and measured BC using light absorption (e.g. Aethalometer, AE) and thermal/optical methods are normally named as "the equivalent BC (eBC)" and "elemental carbon (EC)", respectively. In Table 1, BC concentrations in TP measured by several common techniques were collected and grouped according to the methods to make clearer comparison.*

4.Line 158: The sentence suggests that the intermediate position of the concentration within the TP region may be due to anthropogenic emissions in the surrounding area. However, it does not explain why it is not in a high concentration position.

**Response 4**

Thanks for your comments. The expression here is not very precise. As mentioned in Response 3, it is better to compare the rBC concentration measured by same measuring techniques. Hence, we have modified the description in the Line 178-180 in revised manuscript and further discussed the anthropogenic emissions in the subsequent sections:

*The rBC concentration in Xihai was relatively high compared to southeastern and central TP measured using same technique (Table 1). This was potentially attributed to the strong BC emissions in surrounding area of northeast TP (Fig. 1).*

5.Line 175: Is this difference statistically significant? Are there any indicators for testing the significance of differences that can be reported?

**Response 5**

Thank you very much for reminding us to confirm the significance of the statistical results. The t-test ($\alpha$=0.05, n=51) results proved that the rBC, $PM_{BC}$ and $R_{BC}$ were significantly different between Xihai and Lulang, and we have added the t value in the revised manuscript in Line 194:

*the difference ($t_{rBC}$=2.8, $t_{PMBC}$=2.1) between the two sites was proved by the t-test (α=0.05, v=50).*
and Line 199-200:
*The difference on mixing states of $PM_{BC}$ was also demonstrated by the t-test ($t_{rBC}$=2.4).*

6.Line 192: The sentence could be improved by providing citations to support the claim that
C2H3O+ is a typical biomass burning (BB) tracer.
**Response 6**
Thanks so much for your suggestion, we have added the citations in Line 221-222:
*The POA factor in Lulang had higher fraction of signal of $C_2H_3O^+$ (m/z 60) ion (f_$C_2H_3O^+$) in mass*
*spectrum, which is the fragment of levoglucosan mainly from BB **(Lee et al., 2010)**.*

7.Line 222: 17.3% cannot be reflected in Fig. 5b, but only in Fig. 5a, and there is no particular reason
to switch from two decimal places to one decimal place in the figure.
**Response 7**
Thanks so much for your carefully reviewing, we have corrected this sentence and standardized
decimal places in the revised manuscript:
Line 245: *Besides MO-OOA, $NO_3^-$ (17%) and HOA (35%) al**so made large contribution on BC***
***coating (Fig. 5a) and coated OA (Fig. 5b) in Xihai compared to Lulang.***

8.Line 240: The statement "The abundant NO3- was closely associated with anthropogenic sources"
is mentioned here, but it should be referenced in line 222 to support the conclusion that "indicating that
anthropogenic..."
**Response 8**
Thanks so much for your suggestion. We have modified the sentence following your suggestion to
make the statements appearing in the more appropriate position that is more relevant to the viewpoints:
Line 246-250: ***The HOA and $NO_3^-$ were both closely associated with anthropogenic sources***
***because the anthropogenic sources emitted the HOA (Zhang et al., 2005) and precursors of $NO_3^-$***
***largely (Dall'osto et al., 2009; Richter et al., 2005; Sun et al., 2018).*** *It indicated that anthropogenic*
*emissions have a strong influence on coating process of $PM_{BC}$ in northeast TP, which is quite different*
*from southeast TP.*

**Reference**
Cui, S. J., Huang, D. D., Wu, Y. Z., Wang, J. F., Shen, F. Z., Xian, J. K., Zhang, Y. J., Wang, H. L., Huang, C., Liao,
H., and Ge, X. L.: Chemical properties, sources and size-resolved hygroscopicity of submicron black-carbon-containing
aerosols in urban Shanghai, Atmospheric Chemistry and Physics, 22, 8073-8096, 10.5194/acp-22-8073-2022, 2022.
Dall'Osto, M., Harrison, R. M., Coe, H., Williams, P. I., and Allan, J. D.: Real time chemical characterization of
local and regional nitrate aerosols, Atmospheric Chemistry and Physics, 9, 3709-3720, 10.5194/acp-9-3709-2009, 2009.
Lee, T., Sullivan, A. P., Mack, L., Jimenez, J. L., Kreidenweis, S. M., Onasch, T. B., Worsnop, D. R., Malm, W.,
Wold, C. E., Hao, W. M., and Collett, J. L.: Chemical Smoke Marker Emissions During Flaming and Smoldering Phases
of Laboratory Open Burning of Wildland Fuels, Aerosol Science and Technology, 44, I-V,
10.1080/02786826.2010.499884, 2010.
Richter, A., Burrows, J. P., Nüss, H., Granier, C., and Niemeier, U.: Increase in tropospheric nitrogen dioxide over
China observed from space, Nature, 437, 129-132, 10.1038/nature04092, 2005.
Sun, P., Nie, W., Chi, X., Xie, Y., Huang, X., Xu, Z., Qi, X., Xu, Z., Wang, L., Wang, T., Zhang, Q., and Ding, A.:
Two years of online measurement of fine particulate nitrate in the western Yangtze River Delta: influences of
thermodynamics and N2O5 hydrolysis, Atmospheric Chemistry and Physics, 18, 17177-17190, 10.5194/acp-18-17177-
2018, 2018.

Zhang, Q., Worsnop, D. R., Canagaratna, M. R., and Jimenez, J. L.: Hydrocarbon-like and oxygenated organic
aerosols in Pittsburgh: insights into sources and processes of organic aerosols, Atmospheric Chemistry and Physics, 5,
3289-3311, 10.5194/acp-5-3289-2005, 2005.

---

## Author Comment (AC2)

**Response**

The manuscript provides observational evidence of the spatial discrepancies in the physicochemical properties of black carbon (BC)-containing particles within the Tibetan Plateau region. It emphases the significant impacts of elevated anthropogenic emissions from surrounding low-altitude areas on BC, altering its concentration and chemical composition, as well as enhancing its light absorption ability. The manuscript is well written, and the topic is of interest and fits the scope of ACP. I recommend a minor revision before publication. The detailed comments or suggestions are shown below:

We appreciate the reviewer's kind effort and insightful comments. The amendment and modification have finished followed by all constructive comments in the revised manuscript and supporting information. Please kindly find our point-by-point responses listed below. The reviewer's comments are in blue font followed by our responses and revisions in the manuscript (in Italic).

1.In this study, there are three factors identified using PMF, which is a little less than the usual number (normally 4 or 5 factors can be resolved by PMF for HR-AMS data). It is better to provide the explanation of why the 3-factor result is chosen in the main text or SI.

**Response 1**

Thanks so much for your suggestion. We have added relevant material in SI (Line 43-57, Figure S4-S7) to clarify why the 3-factors solutions were selected.

As the Fig. S4a shows, the reduction rate of PMF was small after the number of factors exceeds 3 and 4, and the measured and reconstructed signals matched well in 3-factors solutions (Fig. S4b). Hence, the 3-factors solution can reasonably analyze the source of OA in Lulang.

[Figure]

*Figure S4:(a) The variation of the reduction rate of Q/Qexp with the number of factors in Lulang. (b) The time series of reconstructed signal and measured signal for 3-factors solution.*

In addition, the Factor 3 in the 4-factors scheme was split primarily from LO-OOA and BBOA. This factor is strongly correlated with all other factors, predicting that this factor is not very independent and representative. It is likely to be a product of over-splitting.

The Factor 3 in the 4-factors scheme is characterized primarily by a strong $C_2H_3O^+$ signal (Fig. S5b), and the $C_2H_3O^+$ signal is usually a tracer ion for fresh SOA. It also has higher correlation to $C_2H_4O_2^+$ (r=0.81), meaning that it may be the aged BBOA. However, the BBOA factors in the 4-factors solution also has a relatively high degree of oxidation compared to fresh BBOA. The meaning of the Factor 3 and
BBOA is faintly repetitive, also indicating that the Factor 3 may the product of over-split.

[Figure]

Figure S5: The mass spectrum of factors in the (a) 3-factors solution and (b) 4-factors solution.

Similar to Lulang, the 3-factors scheme reconstructs the OA concentrations well in Xihai (Fig. S6b).
When the number of factors is high (>4), there is a small decrease in $Q/Q_{exp}$ (Fig. S6a). For the 4-factors
solution, the concentration of the new factor (Factor 4) appeared close to zero for most of the time (Fig.
S7b), indicating that this factor did not represent a stable source of OA. The new Factor 4 only had a brief
increase in concentration during the period of 18-20 June, with basically same variation as the LO-OOA.
Considering above reasons, the Factor 4 is most likely a product of the excessive decomposition.

[Figure]

Figure S6:(a) The variation of the reduction rate of Q/Qexp with the number of factors in Xihai. (b) The time series of reconstructed
signal and measured signal for 3-factors solution.

[Figure]

*Figure S7: The (a) mass spectrum and (b) time series of factors in the 4-factors solution.*

2.Line 131: what refractive index did you use for the core-shell Mie model? Please add the numbers
you adopted and reference here.

**Response 2**

Thanks so much for your reminder, we have added the description of the refractive index we used in
the revised manuscript (Line 145-146): *The refractive index was 1.95 - 0.79i for rBC (Bond and Bergstrom,*
*2006), and was 1.52 - $10^{-6}$i (Pitchford et al., 2007) for BC coating at 550 nm wavelength.*

3.Line 154: It is suggested to clarify the different measuring instruments corresponding to different
BC definitions before comparing BC concentrations in Table 1.

**Response 3**

Thank you for suggesting us to add the necessary clarification on the differences in measurement
methods for black carbon (BC). This suggestion is important because the term "black carbon (BC)" has
not been used rigorously or consistently throughout all previous modeling and measurement literature.
Similar terms including "rBC", "eBC", and "EC" has also been widely used corresponding to different
measurement techniques. We have revised the Table 1, and correspondingly modified the manuscript in
Line 170-176:

*Note that, the term "black carbon (BC)" has not been used rigorously or consistently throughout all*
*previous modelling and measurement literature (Bond et al., 2013). Similar terms including "rBC",*
*"eBC", and "EC" has also been widely used corresponding to different measurement techniques. BC*
*measured by laser-induced techniques is often referred as "rBC", and measured BC using light*
*absorption (e.g. Aethalometer, AE) and thermal/optical methods are normally named as "the equivalent*
*BC (eBC)" and "elemental carbon (EC)", respectively. In Table 1, BC concentrations in TP measured by*
*several common techniques were collected and grouped according to the methods to make clearer*
*comparison.*

4.In Table 1, the description "BC (EC)" is not very precise here. Please revise them.

**Response 4**

Thanks very much for your suggestion, we have corrected this imprecise expression, and the heading of Table 1 has been modified to "BC concentration". We have also added the necessary clarification about the methods and definition of BC as Response 3 mentioned.

5. Table 1: Please add a note illustrating the meaning of the numbers in "BC concentration" column, i.e., is it a mean or median value? What does the range stand for in the parenthesis?

**Response 5**

Thank you for pointing out the lack of data description here. Followed by your advice, the meaning of data has been added in the caption of Table 1 (Line 166-167):

*Table 1: Overview of the BC concentration (mean±1σ) at different sites of TP in existing studies.*

*The minimum value and maximum value were shown in the parenthesis.*

6.Line 106: please add abbreviation of light absorption coefficients ($b_{abs}$) here.

**Response 6**

Thanks so much for your notice, the abbreviation of light absorption coefficients ($b_{abs}$) has been added in Line 157 in revised manuscript:

*$PM_{BC}$ concentration and light absorption coefficients (**$b_{abs}$**) increased in the latter period of Xihai*

*campaign.*

7.Figure 7: the color of CL2 in subplot (b) overlaps with the background map. It is better to change it to ensure the visibility.

**Response 7**

Thanks so much for your advice to improve the visibility of the figure, the color scheme of Fig. 7

has been modified as follow:

[Figure]

*Figure 7: The maps show the backward trajectories in different clusters of (a) Xihai and (b) Lulang. Each circular marker along the*
*trajectories denotes a 24-hour interval. The background shading represents the anthropogenic BC emission intensity and the orange spots*
*represent the location of wildfire during the campaign in (a) and (b). The stacked bar plots show the mass concentration of coating*
*components and rBC in (c) Xihai and (d) Lulang.*

8.Please check and ensure that the number of significant digits are consistent throughout the
manuscript.
**Response 8**
Thanks so much for your advice to improve the uniformity and preciseness of our article. The number
of significant digits has been unified throughout the manuscript, and the modification was highlighted by
blue font in the manuscript.

**Reference**
Bond, T. C. and Bergstrom, R. W.: Light absorption by carbonaceous particles: An investigative review, Aerosol Science
and Technology, 40, 27-67, 10.1080/02786820500421521, 2006.
Bond, T. C., Doherty, S. J., Fahey, D. W., Forster, P. M., Berntsen, T., DeAngelo, B. J., et al.: Bounding the role of black
carbon in the climate system: A scientific assessment. Journal of Geophysical Research: Atmospheres, 118(11), 5380–
5552, 2013.
Pitchford, M., Malm, W., Schichtel, B., Kumar, N., Lowenthal, D., and Hand, J.: Revised algorithm for estimating light
extinction from IMPROVE particle speciation data, Journal of the Air & Waste Management Association, 57, 1326-
1336, 10.3155/1047-3289.57.11.1326, 2007.

---

## Author Comment (AC3)

**Response**

Black carbon (BC) is one of the most important aerosol species affecting climate, glaciers and hydrology in Tibetan Plateau (TP). However, large uncertainties still exist in the estimation of BC DRF over the TP, which is related to the mixing states of BC. This study presents multi-point observations of BC mixing states, especially the chemical composition of BC-containing particles, and combines model simulation to reveal the causes of spatial differences and the impacts of transported emission sources. It provides valuable results which may support the future evaluation of BC climate and environmental effects over the TP region. The coupling mechanism between the planetary boundary layer, topography, and pollution mentioned in the text is also interesting. Overall, it is a well-organized manuscript. Thus, I suggest publishing after minor revisions. The detail comments are shown below:

We appreciate the reviewer's kind effort and insightful comments. The amendment and modification have finished followed by all constructive comments in the revised manuscript and supporting information. Please kindly find our point-by-point responses listed below. The reviewer's comments are in blue font followed by our responses and revisions in the manuscript (in Italic).

1.My main suggestion is that the model configuration and validation need to be more detailed, which can be described in 2.3 part. Additional figures of model validation can be added into the SI (e.g. meteorological parameters, gaseous pollutants). Furthermore, a brief setup of the chemical transport model should be elucidated within the main text, in addition to its inclusion in the Supplementary Information.

**Response 1**

Thank you very much for your advice on the introduction of modelling methods, and we have added more details and validation of the modelling in the main text and the SI.

We have modified the main text as shown below:

1. Line 115-116: *The simulation was conducted for the longer period including the times of whole campaign from 3 June to 11 June 2021*;

2. Line 122-131: *The Yonsei University planetary boundary layer (YSU PBL) scheme was used to parameterize boundary layer processes (Hong et al., 2006). Other essential physical parameterization options included the unified Noah land surface model (Ek et al., 2003), the Lin microphysics scheme (Lin et al., 1983), and the Grell-Freitas cumulus parameterization scheme (Grell and Freitas, 2014). For representing atmospheric chemistry numerically, we utilized the Carbon-Bond Mechanism version Z photochemical mechanism along with the Model for Simulating Aerosol Interactions and Chemistry aerosol module (Zaveri and Peters, 1999; Zaveri et al., 2008). Both natural and anthropogenic emissions were considered in this regional WRF-Chem modeling study. Anthropogenic emissions were derived from the Multi-resolution Emission Inventory for China (MEIC), which includes emissions from power plants, residential combustion, industrial processes, on-road mobile sources, and agricultural activities (Li et al., 2017). Biogenic emissions were calculated online using the Model of Emissions of Gases and Aerosols from Nature (MEGAN), encompassing more than 20 biogenic species (Guenther et al., 2006)*

We have modified the SI in Line 20-32:

[Figure]

*Figure S1: The time series of the near-surface air temperature, sulfur dioxide ($SO_2$), ozone ($O_3$) and mass concentration of fine particulate matter ($PM_{2.5}$) in the Xihai and surrounding area. The line and marker represent the results of ambient measurement and modelling respectively. The MB and NMB are mean bias and normalized mean bias (NMB) of each parameter.*

In this study, the air temperature at 2 m were evaluated based on the measurement data from our measurement and publicly available meteorological datasets of the University of Wyoming (http://www. weather.uwyo.edu/surface/). The air temperature was pretty close between the modelling and measurement, and the mean bias was +1.00 ℃. It was shown that the model had a good performance in the simulation of meteorological fields.

The air quality dataset at Xihai and the monitoring stations near to Xihai (https://quotsoft.net/air/) were used to evaluate the WRF-Chem model in simulating the air pollution. There were overall good agreement and small bias between model-simulated and observed concentration values of gaseous pollutant ($SO_2$, $O_3$) and particulate matter. The modelled $SO_2$ concentration level is relatively low, however, it is not the pollutants of major concern in this study.

2.As I commented earlier in the discussion process, the authors should pay careful attention to the use of significant figures in both the main text and figures. For instance, in Figure 5, the fractions should be reported with 2 significant digits, not 4. Additionally, it is important that the number of significant figures is consistent across all panels in Figures 3 and 7.

**Response 2**

Thanks for your suggestion. The Figure 3, 5 and 7 have been modified, and the number of significant digits (highlighted by blue font in manuscript) has been unified in the manuscript now. The modified figures are shown below:

[Figure]

Figure 3: The box plots of (a) rBC and (b) BC-containing particles mass concentrations in Xihai and Lulang, the lower and upper lines of box plot represent the 25th and 75th percentiles and the whiskers stand for 5th and 95th values. The charts of normalized frequency distribution show (c) mass ratio of coating substance to rBC core ($R_{BC}$) and (d) mass absorption cross-section (MAC). Only 1.15% of the $R_{BC}$ exceeded the maximum value of bin (19.5) in Xihai, and no $R_{BC}$ exceeded the maximum value of bin in Lulang.

[Figure]

Figure 5: The stacked bars represent mass concentrations of (a) different species in BC-containing particles (PM$_{BC}$), and (b) different factors of organic aerosol in BC-containing particles. The numbers on the plot show the percentage of different species and organic factors. In subplot (a), PM$_{BC}$ in the TP (this study) was compared to PM$_{BC}$ in urban regions (Collier et al., 2018; Cui et al., 2022).

[Figure]

Figure 7: The maps show the backward trajectories in different clusters of (a) Xihai and (b) Lulang. Each circular marker along the trajectories denotes a 24-hour interval. The background shading represents the anthropogenic BC emission intensity and the orange spots represent the location of wildfire during the campaign in (a) and (b). The stacked bar plots show the mass concentration of coating components and rBC in (c) Xihai and (d) Lulang.

3.Line 27: the term "self-elevated" is typically associated with plume uplift due to the absorption of solar radiation by black carbon. However, as the uplift in question may not be solely attributed to this mechanism, it is recommended to replace the term.

**Response 3**

Thanks so much for your advice. The relative content has been modified in the Line 26-28 in abstract:

*In contrast to Xihai, the thickly coated BC in Lulang was mainly caused by **elevation and transportation of biomass burning plume** from the South Asia.*

4.Line 36: The expression "2,500,000 km2" appears twice in a sentence. Please remove one instance to ensure clarity and conciseness.

**Response 4**

Thanks so much for pointing out the repetition here. The Line 36 has been modified as followed:

*The Tibetan Plateau (TP) is the largest plateau of the world, covering approximately 2.5 million $km^2$.*

5.Line 150, too many digitals for wind speed and gaseous pollutant concentrations

**Response 5**

Thanks for your suggestion. We have modified the decimal fraction of number in Line 156, 164 and 165.

Line 156: *The mass concentration of rBC shows large temporal variation at both sites, with ranges of **0.02–1.28** $\mu g\ m^{-3}$ in Xihai and **0.02-2.22** $\mu g\ m^{-3}$ in Lulang.*

Line 164-165: *with mean value of **1.8$\pm$1.2** $m\ s^{-1}$ and **1.5$\pm$1.2** $m\ s^{-1}$, respectively. In terms of gaseous pollutants, higher levels of $NO_x$ and $O_3$ were observed in Xihai (**5.3$\pm$3.4 and 48$\pm$13 ppb**) than in Lulang*

*(4.0±2.5 and 35±15 ppb).*

6.Line 175, a t-test is needed when comparing RBC between Xinhan and Lulang

**Response 6**

Thank you very much for reminding us to show the significance of our statistical results. The t-test ($\alpha$=0.05, $\nu$=50) results have proved that the difference of $R_{BC}$ in two sites was significant, and we have modified our manuscript in Line 199-200:

*The difference on mixing states of $PM_{BC}$ was also demonstrated by the t-test ($t_{RBC}$=2.4).*

7.Line 199-200, "MO-OOA had very high O:C (0.84), while the O:C of LO-OOA was only 0.49." It is better to be rephrased to in Xinhai "MO-OOA exhibited higher O:C ratio (0.84) than LO-OOA (0.49) "

**Response 7**

Thanks so much for the suggestion. We have modified our manuscript following your suggestion:

Line 220-221: *MO-OOA exhibited higher O:C ratio (0.84) than LO-OOA (0.49).*

8.Line 201 "Therefore, this POA factor was identified as biomass burning OA (BBOA) in Lulang.".

Since f60 from this factor is not as high as those from biomass burning source test, likely caused by aging process, it is better to state that this factor is likely associated with biomass burning activities.

**Response 8**

Thanks so much for your attention that the BBOA in Lulang was slightly different from the fresher

BBOA. According to your advice, we have added some clarification about BBOA to make it clearer:

Line 223-225: *Moreover, the $f\_CO_2^+$ and $f\_C_2H_4O_2^+$ (0.07 versus 0.025) of this factor were also*

*within the triangle area in previous BBOA study (Cubison et al., 2011), and the $f\_C_2H_4O_2^+$ was lower*

*than the fresh BBOA, indicating that this factor was influenced by biomass burning activities and aging*

*processes collectively.*

9. Figure 6, in the figure caption, it is better to explain how the grid of x-axis is calculated since RBC

equals 0 meaning externally mixed BC.

**Response 9**

Thanks so much for your suggestion. We agree that this may cause the misunderstanding. The $R_{BC}$

was calculated averagely in different bins from 0 to 20 in Xihai, and the width of bin is 1.5. In Lulang, the bin of $R_{BC}$ was from 0 to 12 with the same bin width. The coating composition at $R_{BC}$=0 was average composition of BC coating with $R_{BC}$ between 0 and 1.5 in original figure. We have revised the Fig. 6. The x-axis is corresponding to the median value of each $R_{BC}$ bin, rather than the boundary value of each $R_{BC}$

bin.

[Figure]

Figure 6: The variation of BC coating composition with $R_{BC}$ between (a) Xihai and (b) Lulang. The x-axis represents the mass ratio of BC coating components and rBC cores ($R_{BC}$), and the y-axis represents the mass fractions of BC coating components coated on rBC. **The mass fraction of components was averaged in each bin of $R_{BC}$ (bin width: 1.5).**

10.Line 191: "The POA factor had higher signal of C4H7+ and C4H9+ in its mass spectrum…". Please explain the major sources of C4H7+ and C4H9+ and add proper references here.

**Response 10**

Thanks so much for your reminder, we have explained the major sources of the two important ions and added the references in the revised manuscript.

Line 211-212: *The POA factor had higher signal of $C_4H_7^+$ and $C_4H_9^+$, **which is the important alkyl fragments from primary sources (Hu et al., 2016).***

11. Line 214: "OA was the dominant component of BC coating (Fig. 5b) at both sites". Here it should refer to Fig. 5a rather than Fig. 5b.

**Response 11**

Thanks so much for your carefully reviewing and attention, we have corrected this image number in the revised manuscript.

Line 238: *OA was the dominant component of BC coating **(Fig. 5a)** at both sites.*

12. Line 219: "The dominance of MO-OOA in BC coating was resulted from strong atmospheric oxidizing capacity in TP and fast aging process during transport.". Based on Fig. 5, I don't think there is information about the atmospheric oxidation capacity and aging rates during transport. If this discussion is located in the latter part of the article, it is suggested to place this sentence in a more appropriate location.

**Response 12**

Thank you very much for the advice that makes the figures more relevant to the viewpoints. We have modified the article followed by your suggestion:

Line 242-245: *The BC coating was dominated by MO-OOA which was importantly affected by*

atmospheric oxidizing process. The concentration of $O_3$ highly relative to atmospheric oxidizing capacity improved significantly in afternoon (Fig. S8), and the enhanced oxidizing capacity could cause increase of MO-OOA in BC coating in both Xihai and Lulang.

13. Line 305: is there any observed MAC in TP region? If yes, it is recommended to compare the results in this study with previous observations (references need to be added accordingly).

**Response 13**

Thanks so much, your suggestion is important to demonstrate the representativeness of our observations results. The manuscript has been modified as shown below:

Line 331: *The MAC were all relatively high in three clusters of airmasses of Xihai, with distribution peaked between 12 and 14 $m^2$ $g^{-1}$ **that numerically comparable to previous studies (Wang et al., 2015).***

Line 336: *The MAC in Lulang was also **comparable to previous studies (Wang et al., 2018)** that the peak of MAC distribution was 7.6 $m^2$ $g^{-1}$ at 870 nm (12.0 $m^2$ $g^{-1}$ at 550 nm if the Absorption Ångström Exponent of BC is 1.0).*

Line 338: *In CL1 airmasses of Lulang, MAC mainly distributed at the bin between 12 and 14 $m^2$ $g^{-1}$ that **is close to MAC (13.1 $m^2$ $g^{-1}$ at 550 nm) at other TP sites affected by biomass burning plume (Tan et al., 2021).***

14. The abbreviations of CL 1,2,3 should be defined in the main text.

**Response 14**

Thanks so much for your carefully reviewing, the abbreviations of the three clusters of the airmasses have been clarified in Line 278-279:

*In Xihai, the airmasses were dominantly from eastern region outside of TP, as indicated by **airmasses cluster1 (CL1)**, followed by the airmasses of **cluster2 (CL2)** from the northwest of Xihai, and the airmasses of **cluster3 (CL3)** from west of Xihai (Fig. 7a).*

**Reference**

Ek, M. B.; Mitchell, K. E.; Lin, Y.; Rogers, E.; Grunmann, P.; Koren, V.; Gayno, G. , and Tarpley, J. D.: Implementation of Noah land surface model advances in the National Centers for Environmental Prediction operational mesoscale Eta model, J. Geophys. Res.-Atmos., 108, 16, 10.1029/2002jd003296, 2003.

Grell, G. A. , and Freitas, S. R.: A scale and aerosol aware stochastic convective parameterization for weather and air quality modeling, Atmos. Chem. Phys., 14, 5233-5250, 10.5194/acp-14-5233-2014, 2014.

Guenther, A.; Karl, T.; Harley, P.; Wiedinmyer, C.; Palmer, P. I. , and Geron, C.: Estimates of global terrestrial isoprene emissions using MEGAN (Model of Emissions of Gases and Aerosols from Nature), Atmos. Chem. Phys., 6, 3181-3210, 10.5194/acp-6-3181-2006, 2006.

Hong, S. Y.; Noh, Y. , and Dudhia, J.: A new vertical diffusion package with an explicit treatment of entrainment processes, Mon. Weather Rev., 134, 2318-2341, 10.1175/mwr3199.1, 2006.

Li, M.; Liu, H.; Geng, G. N.; Hong, C. P.; Liu, F.; Song, Y.; Tong, D.; Zheng, B.; Cui, H. Y.; Man, H. Y., et al.: Anthropogenic emission inventories in China: a review, Natl. Sci. Rev., 4, 834-866, 10.1093/nsr/nwx150, 2017.

Lin, Y. L.; Farley, R. D. , and Orville, H. D.: Bulk parameterization of the snow field in a cloud model, Journal of Climate and Applied Meteorology, 22, 1065-1092, 10.1175/1520-0450(1983)022<1065:Bpotsf>2.0.Co;2, 1983.

Zaveri, R. A. , and Peters, L. K.: A new lumped structure photochemical mechanism for large-scale applications, Journal of Geophysical Research: Atmospheres, 104, 30387-30415, 10.1029/1999jd900876, 1999.

Zaveri, R. A.; Easter, R. C.; Fast, J. D. , and Peters, L. K.: Model for Simulating Aerosol Interactions and Chemistry (MOSAIC), J. Geophys. Res.-Atmos., 113, 29, ArtnD1320410.1029/2007jd008782, 2008.

Hu, W. W., Hu, M., Hu, W., Jimenez, J. L., Yuan, B., Chen, W. T., Wang, M., Wu, Y. S., Chen, C., Wang, Z. B., Peng, J. F., Zeng, L. M., and Shao, M.: Chemical composition, sources, and aging process of submicron aerosols in Beijing: Contrast between summer and winter, J Geophys Res-Atmos, 121, 1955-1977, 10.1002/2015jd024020, 2016.

Wang, Q. Y., Huang, R. J., Cao, J. J., Tie, X. X., Ni, H. Y., Zhou, Y. Q., Han, Y. M., Hu, T. F., Zhu, C. S., Feng, T., Li, N., and Li, J. D.: Black carbon aerosol in winter northeastern Qinghai-Tibetan Plateau, China: the source, mixing state and optical property, Atmospheric Chemistry and Physics, 15, 13059-13069, 10.5194/acp-15-13059-2015, 2015.

Wang, Q. Y., Cao, J. J., Han, Y. M., Tian, J., Zhu, C. S., Zhang, Y. G., Zhang, N. N., Shen, Z. X., Ni, H. Y., Zhao, S. Y., and Wu, J. R.: Sources and physicochemical characteristics of black carbon aerosol from the southeastern Tibetan Plateau: internal mixing enhances light absorption, Atmospheric Chemistry and Physics, 18, 4639-4656, 10.5194/acp-18-4639-2018, 2018.

Tan, T. Y., Hu, M., Du, Z. F., Zhao, G., Shang, D. J., Zheng, J., Qin, Y. H., Li, M. R., Wu, Y. S., Zeng, L. M., Guo, S., and Wu, Z. J.: Measurement report: Strong light absorption induced by aged biomass burning black carbon over the southeastern Tibetan Plateau in pre-monsoon season, Atmospheric Chemistry and Physics, 21, 8499-8510, 10.5194/acp-21-8499-2021, 2021.